# Synergistic Antileishmanial Effect of Oregano Essential Oil and Silver Nanoparticles: Mechanisms of Action on *Leishmania amazonensis*

**DOI:** 10.3390/pathogens12050660

**Published:** 2023-04-29

**Authors:** Alex Barbosa Alves, Bruna Taciane da Silva Bortoleti, Fernanda Tomiotto-Pellissier, Ana Flávia Marques Ganaza, Manoela Daiele Gonçalves, Amanda Cristina Machado Carloto, Ana Carolina Jacob Rodrigues, Taylon Felipe Silva, Gerson Nakazato, Renata Katsuko Takayama Kobayashi, Danielle Lazarin-Bidóia, Milena Menegazzo Miranda-Sapla, Idessania Nazareth Costa, Wander Rogério Pavanelli, Ivete Conchon-Costa

**Affiliations:** 1Laboratory of Immunoparasitology of Neglected Diseases and Cancer, Department of Pathological Sciences, Center of Biological Sciences, State University of Londrina, Londrina 86057-970, PR, Brazil; 2Carlos Chagas Institute (ICC-Fiocruz-Pr), Curitiba 81310-020, PR, Brazil; 3Laboratory of Biotransformation and Phytochemistry, Department of Chemistry, Center of Exact Sciences, State University of Londrina, Londrina 86057-970, PR, Brazil; 4Department of Microbiology, Center of Biological Sciences, State University of Londrina, Londrina 86057-970, PR, Brazil

**Keywords:** leishmaniasis, association, cell death, apoptosis-like, ROS, NO

## Abstract

American tegumentary leishmaniasis, a zoonotic disease caused by the *Leishmania* genus, poses significant challenges in treatment, including administration difficulty, low efficacy, and parasite resistance. Novel compounds or associations offer alternative therapies, and natural products such as oregano essential oil (OEO), extracted from *Origanum vulgare*, have been extensively researched due to biological effects, including antibacterial, antifungal, and antiparasitic properties. Silver nanoparticles (AgNp), a nanomaterial with compelling antimicrobial and antiparasitic activity, have been shown to exhibit potent leishmanicidal properties. We evaluated the in vitro effect of OEO and AgNp-Bio association on *L. amazonensis* and the death mechanisms of the parasite involved. Our results demonstrated a synergistic antileishmanial effect of OEO + AgNp on promastigote forms and *L. amazonensis*-infected macrophages, which induced morphological and ultrastructural changes in promastigotes. Subsequently, we investigated the mechanisms underlying parasite death and showed an increase in NO, ROS, mitochondrial depolarization, accumulation of lipid-storage bodies, autophagic vacuoles, phosphatidylserine exposure, and damage to the plasma membrane. Moreover, the association resulted in a reduction in the percentage of infected cells and the number of amastigotes per macrophage. In conclusion, our findings establish that OEO + AgNp elicits a late apoptosis-like mechanism to combat promastigote forms and promotes ROS and NO production in infected macrophages to target intracellular amastigote forms.

## 1. Introduction

*Leishmania amazonensis* is one of the main causative agents of American tegumentary leishmaniasis (ATL), which is a zoonotic disease that affects humans and several species of wild and domestic animals and is characterized by chronic inflammation of the skin and mucous membranes [1]. ATL is transmitted to humans by female sandflies of *Lutzomyia* spp., and its pathogenesis depends on factors such as the virulence of the parasite and the host’s immune response [2].

The current treatment of ATL is based on the elimination of amastigote forms, with pentavalent antimonials (Glucantime^®^ and Pentostam^®^), which have been used since 1920 [3]. However, the available drugs present have low efficiency, side effects, and difficulty in administration [3,4]. In addition, the administration of a single drug has led to the development of strains resistant to available drugs. Thus, the combination of compounds has become a good alternative and has shown advantages over the monotherapy [5].

Natural medicine can provide important treatment alternatives for various diseases, such as leishmaniasis. The in vitro or in vivo anti-leishmanial activity of natural compounds trans-chalcone, *Caryocar coriaceum* extracts, grandiflorenic acid from *Sphagneticola trilobata*, solidagenone from *Solidago chilensis*, and dehydroabietic acid from *Pinus elliottii* were recently described [6,7,8,9,10]. Among the natural products, oregano essential oil (OEO), extracted from the wild plant *Origanum vulgare,* has a range of biological effects studied, such as antiparasitic and antibacterial properties [11,12]. Moreover, OEO has been used in a popular medicine due to its intense effect on skin infection as an antiseptic and skin wound healing [11,13,14,15].

Regarding the leishmanicidal activity of OEO, the in vitro leishmanicidal effect on promastigotes and amastigotes forms of *L. amazonensis* was demonstrated without triggering a proinflammatory response, reducing TNF-α with a high arginase-1/iNOS ratio. The in vivo anti-leishmanial potential of OEO was also demonstrated to be capable of reducing the lesion size in *L. amazonensis*-infected mice [12].

In the same way, silver (Ag) has been used in medicine for at least six millennia because of its efficiency in preventing microbial infections. Silver nanoparticles (AgNp) have been shown as potential antimicrobials, causing disruption of the cell integrity, inhibiting ATP synthesis, enhancing ROS production leading to oxidative stress, and disturbing the process of its replication and cell division [16]. Recently, with the advances in nanotechnology, the production of silver nanoparticles has evolved to biological methods (AgNp-bio), so-called “green synthesis”, becoming less toxic and more cost-effective and eco-friendly, with enhanced bioactivity. This new approach expanded the range of possible therapeutic applications [17,18].

Different therapeutic strategies for the treatment of leishmaniasis patients have been explored, including new natural and synthetic compounds, a combination of compounds, and nanomaterials [19,20]. Therefore, this work aimed to evaluate the anti-leishmania activity of the OEO and AgNp association and the possible morphological, ultrastructural, molecular, and biochemical changes induced by the treatment to *L. amazonensis*, both in free promastigote forms and intra-macrophagic amastigote forms.

## 2. Materials and Methods

### 2.1. Culture of Leishmania (Leishmania) amazonensis

Promastigotes of *L. amazonensis* (MHOM/BR/1989/166MJO) were maintained in culture medium 199 (GIBCO, Invitrogen, New York, NY, USA) supplemented with 10% fetal bovine serum (FBS) (GIBCO, Invitrogen, New York, NY, USA), 1 M HEPES buffer, 1% human urine, 1% L-glutamine, streptomycin and penicillin (GIBCO, Invitrogen), and 10% sodium bicarbonate. The cell culture was maintained in a B.O.D at 25 °C in a 25 cm^2^ culture flask. In all experiments, promastigote forms in the stationary growth phase were used (5-day culture).

### 2.2. Animals and Ethics Committee

The BALB/c mice were kindly supplied by the Carlos Chagas Institute/Fiocruz-PR, Curitiba, Brazil. The animals were kept under sterile conditions in the Vivarium of the Department of Pathological Sciences of the State University of Londrina until they reached an approximate weight of 25–30 g and age between 6–8 weeks, with controlled light and temperature. The study was approved, and the animals were used according to the rules of the Animal Experimentation Ethics Committee of the State University of Londrina (UEL) nº 8595.2018.89.

### 2.3. Oregano Essential Oil

The OEO was obtained from Ferquima Industry and Commerce of Essential Oils (São Paulo, Brazil). This oil (batch 224) was extracted by steam distillation, and its density (0.954 g/mL) and composition (main components: 71% carvacrol, 3% thymol, 4.5% terpinene range, 3.5% paracymene, and 4% beta-caryophyllene) were described in a technical report. A 50% OEO stock solution was prepared in dimethylsulfoxide (DMSO) (Sigma-Aldrich, St. Louis, MO, USA). The maximum concentration of DMSO in the tests was 1%.

### 2.4. Silver Nanoparticles

The biogenic silver nanoparticles (AgNp-bio) were obtained according to Durán et al. (2005a, 2005b) [21,22]. Briefly, the filamentous fungus *Fusarium oxysporum* was grown in a medium containing malt extract (2%) and yeast extract (0.5%) at 28 °C for 6 consecutive days. The grown biomass was filtered and resuspended in sterile water. Approximately 10 g of the fungal suspension was transferred to a conical flask containing 100 mL of distilled water, maintained for 72 h at 28 °C. AgNO_3_ was added (at the final concentration of 10 mM), and the solution was maintained for several hours at 28 °C for the synthesis of the nanoparticles. The recovered silver nanoparticles were characterized through scanning electron microscopy (SEM), X-ray diffraction analysis (XRD), energy dispersive spectroscopy (EDS) demonstrated by Picoli et al. (2016) [18], nanoparticle tracking analysis (NTA) demonstrated by Fanti et al. (2018) [23], the dynamic light scattering (DLS) technique shown in Machado et al. (2020) [24], plasmonic band analysis using the UV–Vis spectroscopy technique, and morphology analysis with transmission electron microscopy (TEM) shown in Sanfelice et al. (2021) [25].

### 2.5. OEO, AgNp, and Association Activity on Promastigote Forms of L. amazonensis

The promastigote forms of *L. amazonensis* (10^6^) were treated with different concentrations of OEO (3.12, 6.25, 12.5, 50, 100 μg/mL), AgNp-bio (0.1, 0.5, 1, 5, 10 μg/mL) and the proportions of the OEO + AgNp associations: 80%/20% (3.2/0.01, 6.4/0.02, 12.8/0.04, 25.6/0.08 μg/mL OEO/AgNp), 60%/40% (2.4/0.02, 4.8/0.04, 9.6/0.08, 19.2/0.16 μg/mL OEO/AgNp), 50%/50% (2/0.02, 4/0.05, 8/0.1, 16/0.2 μg/mL OEO/AgNp), 40%/60% (1.6/0.03, 3.2/0.06, 6.4/0.12, 12.8/0.24 μg/mL OEO/AgNp) ,and 20%/80% (0.8/0.04, 1.6/0.08, 3.2/0.16, 6.4/0.32 μg/mL OEO/AgNp). The parasites were evaluated in a Neubauer chamber after 24 h of treatment. As a negative control, *L. amazonensis* promastigotes kept in an untreated culture medium were used, and as vehicle control, DMSO 0.1% was used.

### 2.6. Peritoneal Macrophage Viability

To evaluate a possible cytotoxic effect of OEO, AgNp-bio, and the associations of both, the MTT assay (3-(4,5-dimethylthiazol-2-yl)-2,5-diphenyltetrazolium bromide) (Sigma-Aldrich) was performed, which measures the metabolic activity of the mitochondria, as described by Mosmann (1983). Macrophages (5 × 10^5^ cells/mL) were recovered from the peritoneal cavity with ice-cold PBS supplemented with 3% FBS and then cultured in 24-well plates with 500 µL of RPMI 1640 medium and 10% FBS for 24 h at 37 °C and 5% CO_2_. The adherent cells were incubated with different concentrations of OEO (16, 32, 64, 128, 256 μg/mL), AgNp-bio (0.2, 0.4, 0.8, 1.6, 3.2 μg/mL), and the proportions of the association: 80%/20% (12.8/0.04, 25./0.08, 51.2/0.16, 102.4/0.32, 204.8/0.64 μg/mL OEO/AgNp-bio), 60%/40% (9.6/0.08, 19.2/0.16, 38.4/0.32, 76.8/0.64, 153.6/1.28 μg/mL OEO/AgNp-bio), 50%/50% (8/0.1, 16/0.2, 32/0.4, 64/0.8, 128/1.6 μg/mL OEO/AgNp-bio), 40%/60% (6.4/0.12, 12.8/0.24, 25.6/0.48, 51.2/0.96, 102.4/1.92 μg/mL OEO/AgNp-bio) and 20%/80% (3.2/0.16, 6.4/0.32, 12.8/0.64, 25.6/1.28, 51.2/2.56 μg/mL OEO/AgNp-bio) for 24 h in the same conditions. Subsequently, the cells were washed with PBS, and MTT (5 mg/mL) was added to the wells, followed by further incubation for 4 h. The MTT product (formazan crystals) was diluted with 300 µL of DMSO, transferred to a 96-well plate, and read on a spectrophotometer (Thermo Scientific, Multiskan GO, Vantaa, Finland) at 550 nm. As a negative control, untreated macrophages were used, while the positive control was cells treated with 4% H_2_O_2_. The results were expressed as a percentage of viability compared to the control group and calculated using the following formula: % (viable macrophages) = sample from the treated group/mean from the untreated control) × 100.

### 2.7. Isobologram Construction Using the Fixed-Ratio Method from OEO and AgNp Combination

The interaction dynamics of AgNp and OEO were studied by the method of fixed proportions described by SEIFERT and CROFT, 2006 [26], to obtain the inhibitory concentration of 50% of the parasites (IC_50_) and cytotoxicity concentration of 50% of the peritoneal macrophages (CC_50_) of each proportion of the OEO/AgNp association. For this, the association of OEO/AgNp was evaluated in the following proportions: (P1 = 80%/20%, P2 = 60%/40%, P3 = 50%/50%, P4 = 40%/60%, and P5 = 20%/80%) that correspond to the doses of (P1 = 12.8/0.04, P2 = 9.6/0.08, P3 = 8/0.1/, P4 = 6.4/0.12, and P5 = 3.2/0.16 μg/mL OEO/AgNp). Thus, was constructed an isobologram for anti-leishmania activity and one for the viability of macrophages, where on the x-axis of the graph the AgNp dose was placed and on the y-axis the OEO dose, drawing an additivity line between the values of the isolated substances, wherein points below the line indicate a synergistic effect, points on the line indicate an additive effect, and points above the line indicate an antagonistic effect described by Zhao, Wientjes, and Au (2004) [27].

### 2.8. Calculation of Combination Index

The value and efficiency of this association can be demonstrated through mathematical calculations using tools such as the combination index (CI) previously described by Hall, Middleton, and Westmacott (1983) [28]. The CI can be calculated using the formula: CI = (IC_50_ combined OEO/IC_50_ OEO alone) + (IC_50_ combined AgNp/IC_50_ AgNp alone) for activity in promastigotes and CI = (CC_50_ OEO combined/CC_50_ OEO alone) + (CC_50_ AgNp combined/CC_50_ AgNp alone) for cytotoxicity on peritoneal macrophages, where CI < 1, =1, and >1 indicate a synergistic, additive, and antagonistic effect described by Chou and Talalay (1984) [29].

### 2.9. Morphological and Ultrastructural Analysis of Promastigotes by Scanning Electron Microscopy (SEM) and Transmission Electron Microscopy (TEM)

Promastigote forms of *L. amazonensis* (10^6^) were treated with the association in the chosen dose (OEO 9.6 μg/mL and AgNp 0.08 μg/mL) and incubated for 24 h. Afterward, the parasites were collected by centrifugation, washed in PBS 0.01 M pH 7.2, and fixed by immersion in 2.5% glutaraldehyde in 0.1 M sodium cacodylate buffer. For evaluation of morphological changes by SEM, promastigotes were fixed with glutaraldehyde adhered to coverslips covered with poly-L-lysine for 60 min. Afterward, they were washed with 0.1 M sodium cacodylate buffer dehydrated in increasing concentrations of ethanol (30–100%), dried at a critical point by replacing ethanol with CO_2_, coated with gold, and analyzed using a high-resolution double beam electron microscope FEI SCIOS.

For evaluation of ultrastructural changes by TEM, promastigotes fixed with glutaraldehyde were transferred to microtubes, washed three times with 0.1 M sodium cacodylate buffer, and post-fixed in a solution of 1% osmium tetroxide, 0.8% potassium ferrocyanide, and 10.0 mM CaCl_2_ in 0.1 M sodium cacodylate buffer at room temperature and protected from light. After, the samples were washed with 0.1 M sodium cacodylate buffer, dehydrated in increasing concentrations of acetone (50–100%), included in EPON resin, and polymerized at 60 °C for 72 h. Ultrathin sections were made in an ultramicrotome, deposited on a copper grid, and contrasted with uranyl acetate and lead citrate for 20 and 10 min, respectively. The analysis was performed using a JEOL JEM 1400 transmission electron microscope.

### 2.10. Detection of Reactive Oxygen Species

Promastigote forms (10^6^) were treated with an association at the chosen dose (OEO 9.6 μg/mL + AgNp 0.08 μg/mL) for 24 h. After, the parasites were washed with PBS and loaded with 10 µM of the probe 2′,7′-dichlorodihydrofluorescein diacetate (H_2_DCFDA) (Sigma-Aldrich) and incubated in the dark for 30 min at 25 °C. As a positive control, 4% H_2_O_2_ was used for 30 min. ROS was measured as an increase in fluorescence caused by the conversion of the non-fluorescent dye to the highly fluorescent 2′,7′-dichlorofluorescein (DCF), in a spectrofluorimeter (Victor X3, PerkinElmer, Turku, Finland), at excitation and emission wavelengths of 488 and 530 nm, respectively.

### 2.11. Determination of Nitrite Levels as an Estimation of Produced Nitric Oxide

Nitrite levels (NO) were determined by the Griess method as an estimation of the nitric oxide produced. Briefly, 60 μL aliquots of supernatants from the anti-promastigote or anti-amastigote assay were recovered, and 60 μL of Griess reagent (1% sulfanilamide and 0.1% of naphthyltylamidine amino acid in orthophosphoric hydrochloride (H_3_PO_4_) 5%) was added. After 10 min of incubation at room temperature, the samples were placed in 96-well microplates. A standard curve was made using serial dilutions of NaNO_2_, and the absorbance was determined at 550 nm in a microplate reader (Thermo Scientific, Multiskan GO).

### 2.12. Determination of Mitochondrial Membrane Potential

Promastigote forms (10^6^) were treated with an association at the chosen dose (OEO 9.6 μg/mL and AgNp 0.08 μg/mL) for 24 h to assess the potential of the internal mitochondrial membrane (ΔΨm). The treated parasites were washed and incubated with 2.5 μM tetramethylrhodamine ethyl ester (TMRE) (Sigma-Aldrich) for 30 min at 25 °C and analyzed in a spectrofluorimeter (Victor X3, PerkinElmer, Finland) at excitation wavelengths of 480 and emission of 580 nm.

### 2.13. Evaluation of Lipid Bodies

Promastigote forms (10^6^) treated with an association at the chosen dose (OEO 9.6 μg/mL + AgNp 0.08 μg/mL) were harvested and washed twice in PBS and stained with 10 μg/mL of Nile red (Sigma-Aldrich) for 30 min at 25 °C. The presence of cytoplasmic lipid bodies was detected in a spectrofluorimeter (Victor X3, Perkin-Elmer, Finland) at excitation and emission wavelengths of 530 and 635 nm, respectively.

### 2.14. Evaluation of Autophagic Vacuoles

The presence of autophagic vacuoles was evaluated in promastigote forms treated for 24 h with the association in the chosen dose (OEO 9.6 μg/mL + AgNp 0.08 μg/mL). The parasites were washed twice in PBS, incubated with 5 μL of monodansilcadaverine (MDC) (Sigma-Aldrich) for 1 h at 25 °C, and analyzed using a spectrofluorimeter (Victor X3, Perkin-Elmer, Finland) at excitation and emission wavelengths of 380 and 525 nm, respectively.

### 2.15. Detection of Death Mechanisms

Promastigote forms (10^6^) treated with the combination at the chosen dose (OEO 9.6 μg/mL + AgNp 0.08 μg/mL) for 24 h were washed and resuspended in 100 μL of 1x assay buffer (Santa Cruz Biotechnology, Santa Cruz, California, USA), followed by adding a mixture containing 1 μL of annexin V/FITC (Invitrogen, Eugene, Oregon, USA) and 1 μL of propidium iodide (PI) (Santa Cruz Biotechnology). Data analysis was performed using a BD Accuri™ C6 Plus flow cytometer. A total of 10,000 events were acquired. Cells negative for annexin V and PI were considered viable, cells stained with annexin V (positive or negative for PI) were considered apoptotic, and cells positive for PI (and negative for annexin V) were classified as necrotic [8]. The cell size was also assessed by flow cytometry in parasites under the same conditions. FSC-A represented the cell volume, and a total of 10,000 events were acquired [7].

### 2.16. Anti-Amastigote Assay

Peritoneal macrophages from BALB/c mice (5 × 10^5^) were cultured in 24-well plates containing glass coverslips, incubated with 500 μL of RPMI 1640 medium for 24 h at 37 °C and 5% CO_2_. The adhering macrophages were infected with promastigote forms of *L. amazonensis* (2.5 × 10^6^) for 2 h. After infection, the non-internalized promastigote forms were removed by washing with PBS, and the adherent cells were treated with the combination of the compounds of the chosen proportion (OEO 9.6 μg/mL and AgNp 0.08 μg/mL), RPMI 1640 medium (negative control), 0.1% DMSO (vehicle), amphotericin B 1 µM (positive control) for 24 h at 37 °C and 5% CO_2_. Then, the cells were stained with methylene blue eosin solution according to Leishman (Leishman’s dye) (Inlab, São Paulo, SP, Brazil). A total of 20 fields were analyzed by immersion at 1000x magnification using an optical microscope (Olympus BX41, Olympus Optical Co., Ltd., Tokyo, Japan).

### 2.17. Production of ROS in Macrophages Infected with L. amazonensis

Macrophages infected with amastigote forms were treated with the dose of the chosen proportion (OEO 9.6 μg/mL + AgNp 0.08 μg/mL) for 24 h. After this period, the cells were washed with PBS and loaded with 10 µM of H_2_DCFDA and incubated for 30 min at 37 °C. As a positive control, 4% H_2_O_2_ was used for 30 min. ROS were measured as an increase in fluorescence caused by the conversion of the non-fluorescent dye to the highly fluorescent DCF in a spectrofluorometer (Victor X3, PerkinElmer, Finland) at excitation and emission wavelengths of 488 and 530 nm, respectively.

### 2.18. Statistical Analysis

The statistical analyzes were determined by ANOVA, followed by the Tukey test for multiple comparisons. Three independent experiments were carried out, each with datasets in triplicate. The data were expressed as mean ± standard error of the mean. The data were analyzed using GraphPad Prism statistical software (GraphPad Software, Inc., San Diego, CA, USA, 500–288). *p*-value ≤ 0.05 was considered statistically significant.

## 3. Results

### 3.1. OEO and AgNp Exert a Leishmanicidal Effect on Promastigote Forms of L. amazonensis and in Combination Present a Synergic Effect with Better Results and Low Toxicity in Peritoneal Macrophages of BALB/c Mice

After 24 h of treatment with different doses of OEO (3.12–100 µg/mL) and AgNp (0.1–10 µg/mL), we verified that the compounds reduced the viability of promastigote forms by 7.61, 36.11, 46.37, 54.03, 86.7, 98.11% for 3.12, 6.25, 12.5, 25, 50, 100 µg/mL of OEO and 33.87, 61.56, 96.95, 100 and 100% for 0.1, 0.5, 1.0, 5.0, 10.0 µg/mL of AgNp, respectively. From these results, it was possible to calculate the half maximal inhibitory concentration (IC_50_) values of the separate compounds, which were the concentration of 16.0 ± 0.05 μg/mL to OEO and 0.2 ± 0.06 μg/mL to AgNp (Figure 1A,B).

In order to reduce the concentrations used of each of the compounds, aiming at a reduction in cytotoxicity and a concomitant increase in the leishmanicidal potential, we analyzed the association of the tested compounds in different proportions. These proportions were tested on promastigote forms, and the IC_50_ values of each proportion were calculated. The obtained values of IC_50_ of each proportion were P1 = 12.7/0.04 (±0.03), P2 = 7.4/0.06 (±0.04), P3 = 8.0/0.10 (±0.02), P4 = 5.1/0.09 (±0.03), and P5 = 2.5/0.13 (±0.03) OEO/AgNp μg/mL (Table 1). Then, the IC_50_ values of the proportions were used to calculate the combination index (CI), and the results showed that the proportions P2 = 60%/40%, P4 = 40%/60%, and P5 = 20%/80% of OEO/AgNp presented a CI > 1 (Table 2), demonstrating a synergistic effect. These results have also been demonstrated through the isobologram, which showed that these proportions are below the line of additivity (Figure 2A). On the other hand, the proportions P1 = 80%/20% and P3 = 50%/50% of OEO/AgNp presented a CI = 1, indicating an additive effect.

Knowing that the treatments have an effect on promastigote forms, we studied their effects on the viability of macrophages, the main host cells for *Leishmania* parasites [30]. We found that the treatment of isolated compounds on peritoneal macrophages showed the half maximal cytotoxic concentration CC_50_ of 116.50 ± 0.04 μg/mL and 2.25 ± 0.09 μg/mL for OEO and AgNp, respectively. Subsequently, all proportions were assessed by obtaining the CC_50_ values (P1 = 105.6/0.33 (±0.08), P2 = 90.48/0.75 (±0.02), P3 = 105.90/1.32 (±0.03), P4 = 82.01/1.53 (±0.02), and P5 = 41.06/2.05 (±0.03) μg/mL OEO/AgNp), and the CI value was calculated, with antagonistic results showing CI > 1 (Table 2). The antagonistic characteristic of the association was also observed in the isobologram, where all proportions with 50% of toxicity are above the tendency line of the graph (Figure 2B).

In the host cell, the treatment proved to be less toxic, since higher doses were necessary to obtain CC_50_. From these results, we selected the fixed ratio P2 = 60%/40% of OEO/AgNp dose 9.6/0.08 μg/mL, since this combination presented the lowest CI, 0.76, for the subsequent experiments.

### 3.2. Combination of OEO and AgNp Induces Morphological and Ultrastructural Alterations in L. amazonensis Promastigotes

After determining the best proportion (P2), SEM and TEM were performed to determine the morphological and ultrastructural changes induced by the treatment in promastigote forms. Untreated parasites showed normal characteristics compatible with an elongated body, flagellum proportional to body size, smooth and intact cell surface, and well-preserved structures (Figure 3A,E). Meanwhile, parasites treated with the association showed morphological and ultrastructural changes, such as cell surface roughness, rounded shape and reduction in cell body size, reduced flagellum, leakage of cytoplasmatic contents (Figure 3B–D), mitochondrial swelling, accumulation of lipid-storage bodies and autophagic vacuoles in the cytoplasm, DNA disorganization in nuclei, and damage to the plasma membrane (Figure 3F–H).

### 3.3. Combination of OEO and AgNp Exerts an Anti-Promastigote Effect by ROS and NO Production Resulting in Mitochondrial Dysfunction, Forming Lipid Bodies and Autophagy Vacuoles

Based on previous results, we decided to evaluate the production of ROS and NO, the main microbial molecules, in OEO-AgNp-treated parasites. Our results demonstrated that the association induced an increase in both ROS and NO in promastigotes when compared with the control (Figure 4A,B).

As TEM analysis indicated mitochondrial damage in treated parasites, we also aimed to confirm that parasite mitochondria were affected by the treatment. As expected, we found a decreased total TMRE fluorescence of 21.7% in the treated group when compared to the control, indicating a loss of mitochondrial membrane potential (Figure 4C).

Analysis by TEM also showed an accumulation of lipid-storage bodies in the cytoplasm; thus, we investigated their presence in parasites treated with the association by staining with Nile red, a fluorescent dye with a high affinity to neutral lipids [31]. Our results exhibited an increase in fluorescence by 76.68% when compared to the control, showing an increase in the formation of lipid bodies (Figure 4D).

Additionally, it was observed by TEM that treatment with association triggered the intense formation of autophagic vacuoles in promastigotes. Thus, we also evaluated whether the treatment induced an increase in autophagic vacuoles, which may precede cell death. Parasites treated and marked with MDC presented an increase in fluorescence of 180% concerning the control, suggesting the formation of autophagy vacuoles (Figure 4E).

### 3.4. The Treatment with OEO + NpAg Culminates in Late Apoptosis in Promastigote Forms

Knowing that OEO + NpAg treatment induces ROS and NO production, which leads to mitochondrial dysfunction, lipidic bodies storage, and autophagic vacuole formation, we aimed to identify the type of death triggered by this cascade of processes. For this, we performed the co-labeling of annexin V and propidium iodide (PI). Our data showed that the proportion of viable parasites decreases in OEO + NpAg-treated promastigotes, while the annexin V+ and annexin V+/PI+ subpopulation enhanced significantly when compared to the control (Figure 5A,B), indicating that most of the treated parasites were in apoptosis/late apoptosis death.

We also evaluated the promastigote size, where we found that the OEO + NpAg treatment was able to significantly reduce the size of parasites when compared to the control (Figure 5C,D), confirming the results found by SEM.

### 3.5. Combination of OEO and AgNp Induces NO and ROS Increase in Infected Macrophages and Anti-Amastigote Activity

Since amastigotes are intracellular forms, we investigated the effect of the association on infected macrophages. For this, the production of NO and ROS in these cells was evaluated. The results showed a significant increase in NO and ROS compared to the control (Figure 6).

Consequently, in infected and treated macrophages, we found a significant decrease in the percentage of infected cells, with a reduction of 34% ± 2.96 (*p* ≤ 0.0001), similar to the positive control, AmB, which induced a reduction of the 46% ± 1.39 (*p* ≤ 0.0001) when compared to the control. With regard to the number of amastigotes per macrophage, the association has also shown a similar effect to the standard drug (AmB), presenting a reduction of 30.7% (*p* ≤ 0.01) (Figure 7).

## 4. Discussion

The current treatment of leishmaniasis has some flaws, such as difficulty in administration, toxicity leading to adverse effects, and parasite resistance, which drives us to search for new treatment strategies for leishmaniasis [32]. In this sense, the search for new, active, natural compounds that are more effective against *Leishmania* spp. parasites, without side effects, is increasing. OEO exhibited a potential leishmanicidal effect on *L. amazonensis* promastigote forms, resulting in a combination of autophagic, apoptotic, and necrotic events [12].

The use of nanotechnology has also shown to be a promising strategy, since it is a delivery system that carries drugs to specific targets [19,23,33]. Moreover, the association of drugs is also a promising method that has stood out in the research field of new drugs against this disease [21,34,35,36].

In our study, an important synergistic leishmanicidal effect of the association of OEO + AgNp was demonstrated, since some proportions showed CI values < 1, and in the isobologram, some proportions were below the additivity line. Similar results were observed by Scandorieiro et al. (2016) [36], which showed the synergetic effect of the association of OEO + AgNp in multi-resistance bacteria. An antagonist effect of the association was demonstrated in peritoneal macrophages, indicating improved safety in the administration of this association regarding the isolated products. Pastor et al. (2015) [35] also presented similar results using the combination of ascaridol and carvacrol with CI > 1 and combinations above the additivity line in isobolograms.

SEM and TEM showed that the association induced several morphological and ultrastructural changes in promastigotes of *L. amazonensis*. Previous studies using different natural or synthetic compounds on *L. amazonensis* promastigotes reported similar changes to those observed in our study [6,7,8,9,10,12].

We also showed the likely mechanism by which this association exerts its antileishmanial effects. We initially focused our studies on investigating alterations in mitochondria. Notable characteristics of *Leishmania* spp. are its single mitochondria, and the survival of this parasite depends on the perfect functioning of this organelle [37].

We observed an increase in the production of ROS in treated promastigotes. ROS are molecules derived from the incomplete one-electron reduction in molecular oxygen, and high concentrations might induce oxidative damage [38]. The association of OEO + AgNp also increases the NO, a short-lived free radical, then can react with proteins and nucleic acids, and we also know the high on NO levels is related to wound healing [39]. Therefore, this association may be inducing an oxidative imbalance in promastigotes, attributable to an increase in oxidant species. Similar results were obtained previously in trypanosomatids by using other active compounds [9,39].

Increases in oxidant species induce mitochondrial dysfunction, such as mitochondrial swelling and alterations in the mitochondrial membrane potential ΔΨm [40]. In this study, TEM showed mitochondrial swelling, and TMRE staining revealed marked reductions in the ΔΨm. The ΔΨm is essential for maintaining the physiological function of the respiratory chain, and a significant loss of ΔΨm depletes cells of energy, with subsequent death [37]. Studies of leishmanicidal compounds that target *L. amazonensis* mitochondria have been published, showing changes in ΔΨm [7,8,10,41].

Since mitochondrial dysfunctions, such as mitochondrial depolarization, induce an increase in lipid bodies, the association of OEO + AgNp also increased lipid droplets formation in the cytoplasm, revealed here by the Nile red and TEM [31]. Antinarelli et al. (2018) [42] also found characteristics of mitochondrial dysfunction to be associated with an increase in lipid bodies when promastigotes of *L. amazonensis* were exposed to 4-hydrazinoquinoline analog, these results being compatible with those found in our study.

The mitochondrial depolarization associated with the accumulation of lipid bodies in the cytoplasm is an event characteristic of the occurrence of apoptosis-like cell death. It is well described in the literature that apoptosis is characterized by several morphological features, among which are cell rounding, cell shrinkage, chromatin condensation, nuclear fragmentation, ultrastructural modifications of cytoplasmic organelles, and plasma membrane modifications with the maintenance of its integrity membrane [43,44]. Our study observed that OEO + AgNp induced similar changes in promastigotes of *L. amazonensis*, revealed by electron microscopy, and all of the morphological, ultrastructural, and biochemical alterations induced by the association suggest apoptosis-like cell death.

In this way, we used phosphatidylserine, a phospholipid naturally present in the inner plasma membrane leaflet that switches to the outer leaflet during apoptosis; annexin V, a marker that binds in phospholipid classes [43]; and PI, a probe that binds to DNA only in ruptured membrane cells [8] to investigate the mechanism of death and found that the treatment with the association demonstrated positive parasites for annexin V/FITC and PI co-staining, indicating late apoptosis.

The presence of autophagy vacuoles was also observed in our study by MDC labeling. The increase in the formation of autophagic vacuoles can be mainly induced by the increases in cellular ROS levels [7,31,38]. Autophagy can induce apoptosis by degrading parts of the cell after sequestration in autophagosomes and degradation within lysosomes or by activating the apoptotic pathway [31,42].

Amastigotes are obligatory intracellular parasites localized in parasitophorous vacuoles of phagocytic cells of the host [2]. AgNp-Bio [23] and OEO exhibited a leishmanicidal effect on intracellular *L. amazonensis* amastigotes forms and reduced the number of infected macrophages [12].

In our study, we observed the anti-amastigote effect of the OEO + AgNp association in macrophages infected by *L. amazonensis*, since a reduction in the percentage of infected macrophages and the number of intracellular parasites was observed. An increase in ROS and NO in infected macrophages was also observed after treatment with the association. Several studies also describe similar results, demonstrating that the probable mechanism of death of *Leishmania* spp. is through molecules such as ROS and NO released by infected macrophages [7,9,10].

However, the increase in ROS and NO in infected macrophages with amastigotes of *L. amazonensis* was not observed by Fanti et al. (2018) [23], which used only AgNp, which leads us to believe that this increase found in our study is due to the presence of OEO and/or the synergistic effect produced by the association of OEO + AgNp. Combination therapy has been used on different models to increase the best action of each compound [34,35,36] and provides a new and innovative approach for future leishmaniasis studies.

## 5. Conclusions

Our research findings reveal a strikingly synergistic leishmanicidal effect in vitro on promastigote forms of *L. amazonensis* through the association of OEO and AgNp, accomplished via metabolic, morphologic, and structural events that induce late apoptosis. In addition, this combination exhibited a remarkable reduction in cytotoxicity in peritoneal macrophages while concurrently eliciting a potent leishmanicidal effect in intracellular amastigotes, possibly via the induction of ROS and NO. These results serve as a powerful impetus to continue our efforts in unraveling the intricate mechanisms of action underpinning the death of this parasite. This combination therapy study determined the mechanisms involved in in vitro OEO + AgNp association against *L. amazonensis*-infected macrophages and highlights a new and innovative therapeutic approach that can be explored as a candidate for future investigations in drug development in leishmaniasis treatment.

## Figures and Tables

**Figure 1 pathogens-12-00660-f001:**
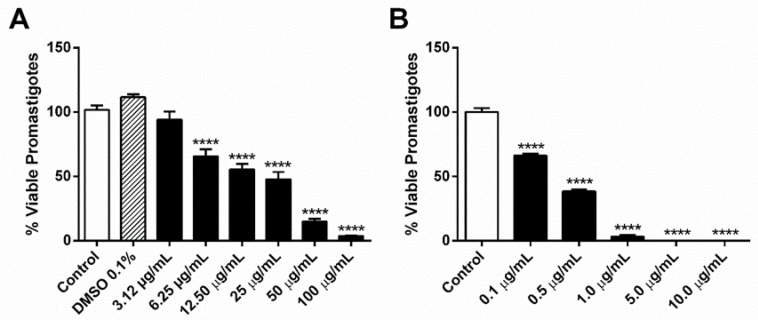
Effect of OEO and AgNp on *L. amazonensis* promastigotes treated for 24 h. (**A**) OEO (3.125, 6.25, 12.5, 25, 50, 100 μg/mL) and (**B**) AgNp (0.1, 0.5, 1, 5, 10 μg/mL) percentage of viable promastigotes. DMSO 0.1% was used as vehicle control. The values represent the mean ± standard error of the mean (SEM) of three independent experiments. Asterisks indicate a significant difference compared to control **** *p* ≤ 0.0001.

**Figure 2 pathogens-12-00660-f002:**
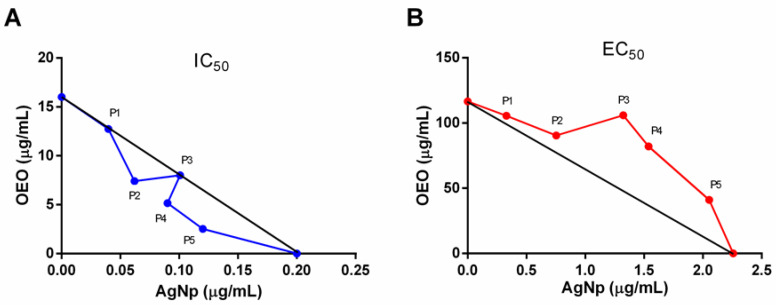
Representative isobologram of in vitro interactions between OEO and AgNp using the method of fixed ratios. Numbers on the axes represent doses calculated from the IC_50_ and CC_50_ values of each compound. (**A**) Activity on promastigotes of *L. amazonensis*; (**B**) cytotoxicity on peritoneal macrophages from BALB/c.

**Figure 3 pathogens-12-00660-f003:**
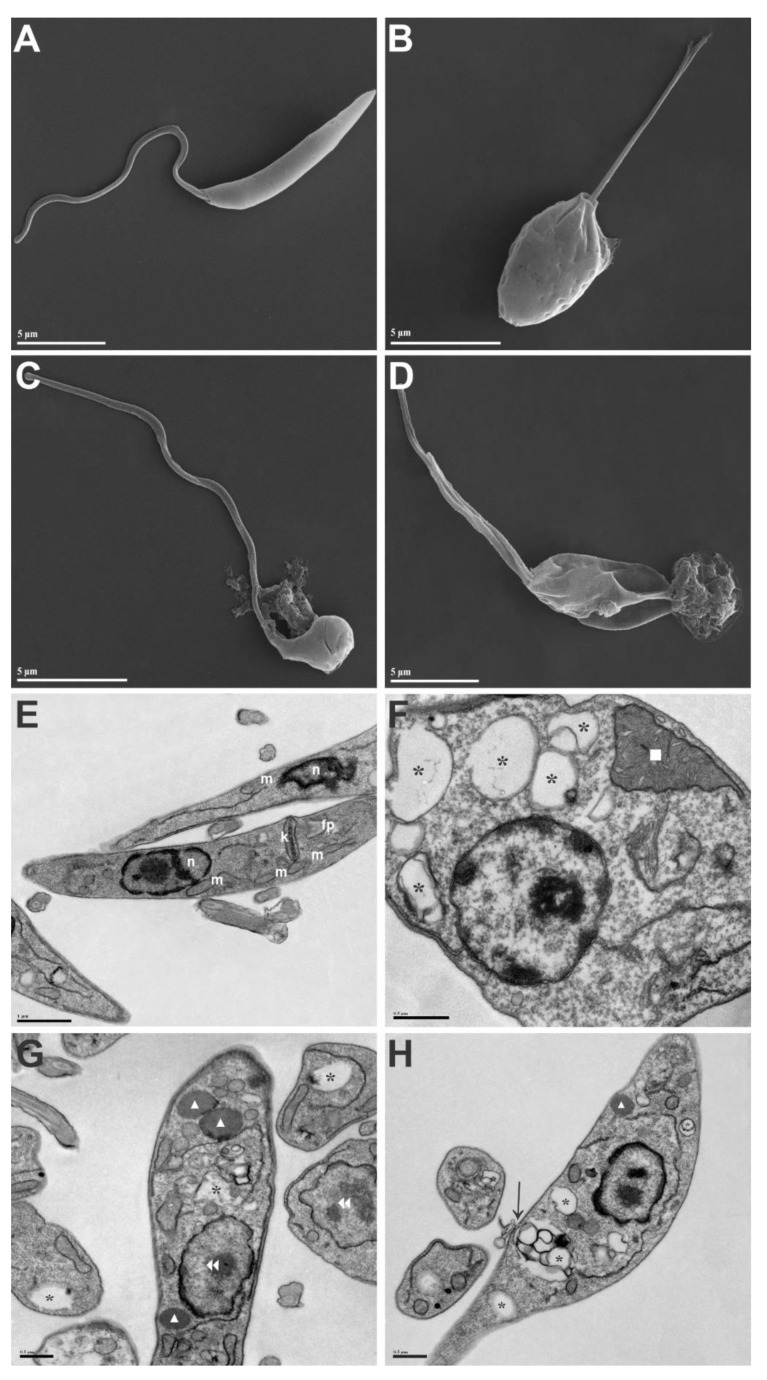
Morphological and ultrastructural changes on promastigotes of *L. amazonensis* treated with 9.6/0.08 μg/mL OEO/AgNp for 24 h. (**A**–**D**) SEM images: (**A**) untreated promastigotes; (**B**–**D**) OEO/AgNp treated-promastigotes. (**E**–**H**) TEM images: (**E**) untreated promastigotes; (**F**–**H**) OEO/AgNp treated-promastigotes. fp, flagellar pocket; k, kinetoplast; m, mitochondrion; n, nucleus; *, autophagic vacuole; ■, mitochondrial swelling; ▲, lipid-storage bodies; 
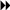
, DNA disorganization in nuclei; →, alteration of the plasma membrane. Scale bars = 5 μm (**A**–**D**), 1 μm (**E**), 0.5 μm (**F**–**H**).

**Figure 4 pathogens-12-00660-f004:**
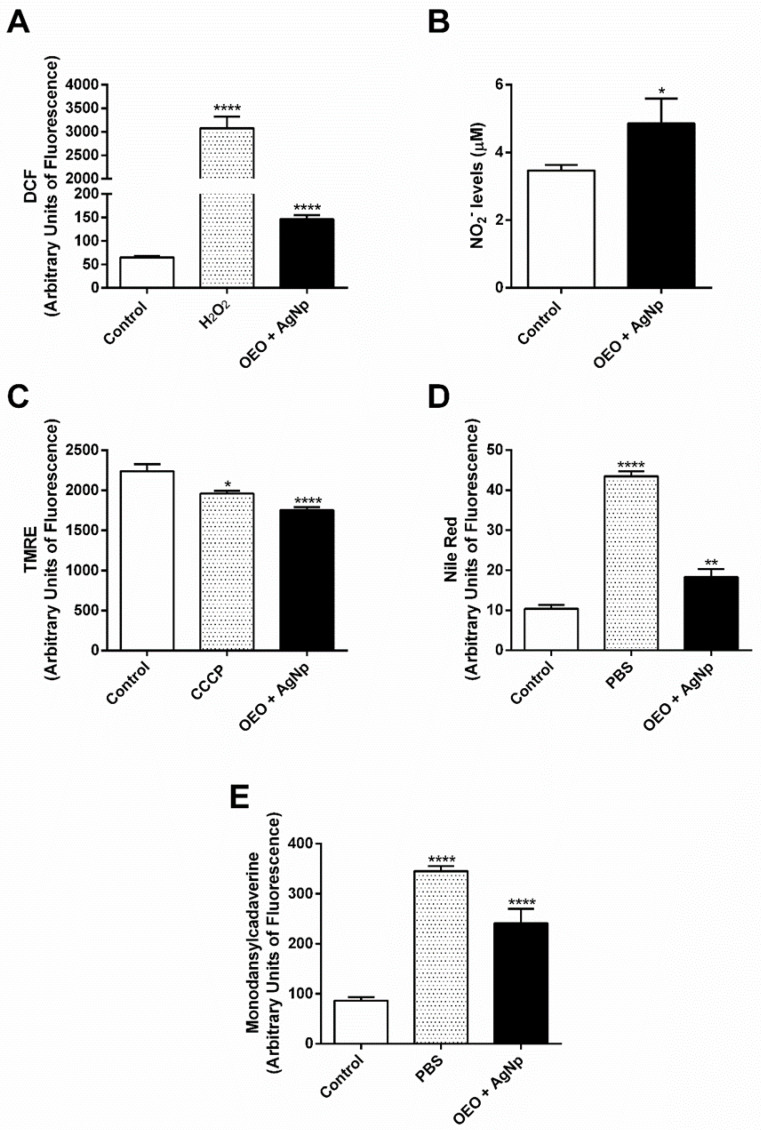
Biochemical targets affected by treatment with the association of OEO/AgNp in *L. amazonensis* promastigotes treated with the dose combination of 9.6/0.08 μg/mL OEO/AgNp. (**A**) Total ROS were measured by an increase in fluorescence caused by the conversion of nonfluorescent dye, H_2_DCFDA, to fluorescent, DCF; (**B**) Griess method for nitrite levels in supernatant; (**C**) mitochondrial membrane potential assay using TMRE labeling; (**D**) lipid-storage bodies using Nile red labeling; (**E**) autophagy vacuoles using MDC labeling. Data represent the mean ± SEM of three independent experiments performed in duplicate. Asterisks indicate significant difference compared to control * (*p* ≤ 0.05), ** (*p* ≤ 0.01), **** (*p* ≤ 0.0001).

**Figure 5 pathogens-12-00660-f005:**
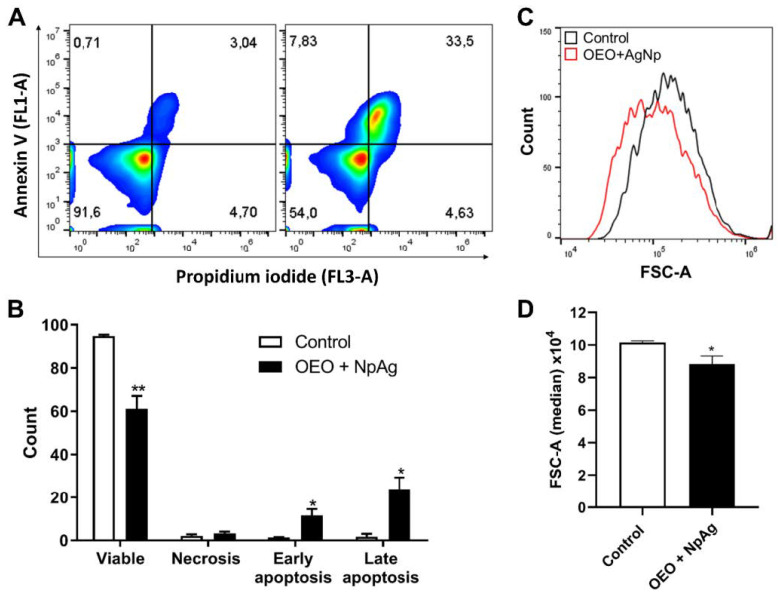
Phosphatidylserine exposure in promastigotes treated with the association at the chosen dose (9.6/0.08 μg/mL OEO/AgNp) using annexin V/FITC and PI. Data acquisition and analysis flow cytometry with a total of 10,000 events. (**A**) Typical dot plot of three independent experiments. (**B**) Count of *L. amazonensis* promastigotes co-staining of annexin V/FITC and PI analyzed by flow cytometry. Asterisks indicate significant difference compared to control * (*p* ≤ 0.05), ** (*p* ≤ 0.01). (**C**) Cell size of *L. amazonensis* promastigotes treated with OEO + NpAg and (**D**) quantitative analysis of cell size. FSC-A was considered a function of cell size. The black line corresponds to the control (untreated parasites), and the red area indicates the parasites treated with OEO + NpAg.

**Figure 6 pathogens-12-00660-f006:**
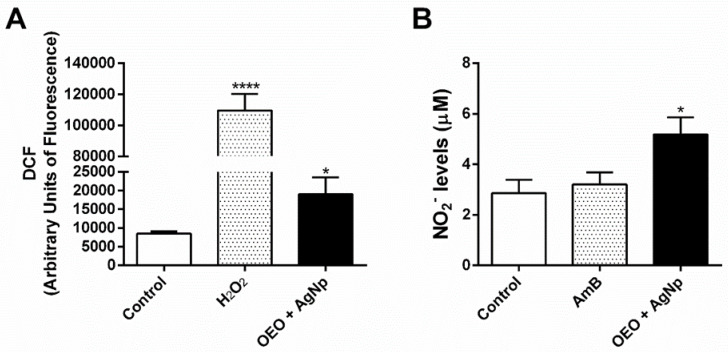
Intramacrophagic *L. amazonensis* death by the association of OEO/AgNp is dependent on NO and ROS. *L. amazonensis*-infected macrophages submitted to a 24 h of treatment with OEO/AgNp (**A**) fluorescent probe DCF for reactive oxygen species measurement in culture cells, (**B**) Griess method for nitrite levels in the culture supernatant. The values represent the mean ± SEM of three independent experiments performed in duplicate. * (*p* < 0,05), **** (*p* < 0,0001).

**Figure 7 pathogens-12-00660-f007:**
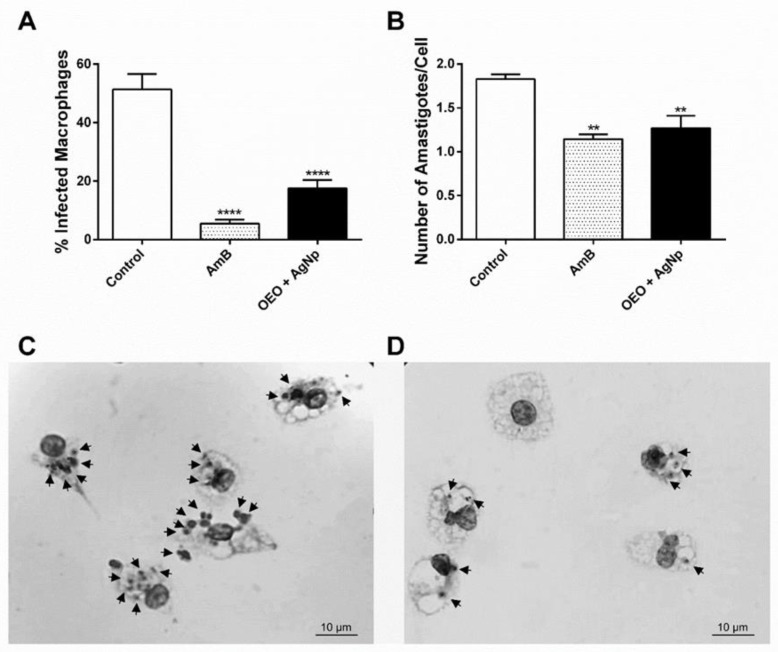
Effect with the association of OEO/AgNp on *L. amazonensis*-infected macrophages. *L. amazonensis*-infected macrophages were treated with the association of OEO/AgNp for 24 h, and the percentage of infected macrophages (**A**) and the number of amastigotes per macrophage (**B**) were evaluated. AmB 1µM (positive control). The values represent the mean ± SEM of three independent experiments performed in duplicate. Significant difference compared to control ** (*p* ≤ 0.01) **** (*p* ≤ 0.0001) photomicrographs of infected macrophages (**C**) and macrophages infected and treated with the association (**D**) stained with Leishman dye under 1000x magnification in an optical microscope. Arrows indicate amastigote forms. Scale bar = 10 µm.

**Table 1 pathogens-12-00660-t001:** Proportion between the amounts of OEO/AgNp using the IC_50_ values of the substances alone.

	Fixed Ratio (%) OEO/AgNp	IC_50_ (µg/mL)	CC_50_ (µg/mL)
OEO	100/0	16 (±0.05)	116.50 (± 0.04)
P1	80/20	12.7/0.04 (±0.03)	105.60/0.33 (±0.08)
P2	60/40	7.4/0.06 (±0.04)	90.48/0.75 (±0.02)
P3	50/50	8/0.10 (±0.02)	105.90/1.32 (±0.03)
P4	40/60	5.15/0.09 (±0.03)	82.01/1.53 (±0.02)
P5	20/80	2.52/0.13 (±0.03)	41.06/2.05 (±0.03)
AgNp	0/100	0.2 (±0.06)	2.25 (±0.09)
AMB		0.06 (±0.02)	45.94 (±0.04)

IC_50_: dose required to eliminate 50% of promastigote forms, CC_50_: dose needed to eliminate 50% of macrophages in the cytotoxicity assay, AMB: amphotericin B.

**Table 2 pathogens-12-00660-t002:** Combination index (CI).

	Fixed Ratio (%) OEO/AgNp	CI IC_50_	CI CC_50_
P1	80/20	0.99	1.05
P2	60/40	0.76	1.11
P3	50/50	1	1.49
P4	40/60	0.77	1.38
P5	20/80	0.80	1.26

CI = 1: the combination is additive, CI < 1: the combination is synergistic, CI > 1: the combination is antagonistic.

## Data Availability

The data presented in this study are available upon reasonable request from the corresponding author.

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
