# Peer review of "Synergistic Antileishmanial Effect of Oregano Essential Oil and Silver Nanoparticles: Mechanisms of Action on Leishmania amazonensis"

_pathogens, 2023, doi:10.3390/pathogens12050660_

Round 1

Reviewer 1 Report

I reviewed the article and considered it a technically well-conducted work with interesting initial results on a potential compound for the treatment of leishmaniasis using silver nanoparticles and oregano essential oil. With the exception of a small correction in the caption of figure 3. 

Figure legend 3 – Include the symbol used to show disorganization of DNA into nuclei

Reviewer 2 Report

1- Why did auteurs use stationary phase of promastigote instead of log phase? (Line 79)

2- In line78, CMneed to edit to superscript. There are a few other similar corrections like line 118,... .

3- 40%/60% and 60%/40% OEO/ AgNp proportions both had CI>1 and 50%/50% OEO/ AgNp had CI=1. How do auteurs describe this condition?

4- Auteurs didn't have any positive control (using meglomin antimonates as a drug choice or  Amphotericin B) to compare how effective this combination would be on promastigotes.

5- Auteurs didn't mention how many replications they had  for OEO, AgNp, and association activity on promastigote forms of L. amazonensis.

6- Auteurs would be better to conclude the structural effects of this combination on the L. amazonensis as well.

7-  In the introduction, auteurs didn't mention why they used biogenic silver nanoparticles instead of chemical ones? Are they more effective or easier to produce?

Reviewer 3 Report

Manuscript review "Synergistic Antileishmanial Effect of Oregano Essential Oil and Silver Nanoparticles: Mechanisms of Action on Leishmania amazonensis" written by Alex Barbosa Alves, Bruna Taciane da Silva Bortoleti, Fernanda Tomiotto-Pellissier, Ana Flávia Marques Ganaza, Manoela Daiele Gonçalves, Amanda Cristina Machado Carloto, Ana Carolina Jacob Rodrigues, Taylon Felipe da Silva, Gerson Nakazato, Renata Katsuko Takayama Kobayashi, Danielle Lazarin-Bidóia, Milena Menegazzo Miranda-Sapla, Idessania Nazareth Costa, Wander Rogério Pavanelli, Ivete Conchon-Costa. The manuscript is devoted to an important problem. The manuscript is structured, written clearly and understandably. Read with interest. However, the manuscript has two minor issues that the authors will need to address.
1. The introduction section of the manuscript is written in an interesting way, but has no obvious connection with the subject of the manuscript. Usually, the introduction sets out already known facts and poses questions that need to be answered. In this case, it is not clear why silver nanoparticles have antibacterial properties? How are these properties implemented? Why is green synthesis better? The authors needs to answer to these questions in the introduction. For convenience, I suggest that the authors use the review article on this topic (10.3390/ph15080968).
2. Nanoparticles not characterized. It is not clear what kind of nanoparticles is being studied. Authors need to characterize their nanomaterials. In the minimum case, authors need to at least know the size of the nanoparticles to be sure that we are dealing with nanoparticles. In a good variant, it is necessary to do DLS, TEM, FTIR, EDX (or other method for determining the chemical composition).

Round 2

Reviewer 3 Report

Ok